# Study of Five-Hundred-Meter Aperture Spherical Telescope Feed Cabin Time-Series Prediction Studies Based on Long Short-Term Memory–Self-Attention

**DOI:** 10.3390/s24216857

**Published:** 2024-10-25

**Authors:** Shuai Peng, Minghui Li, Benning Song, Dongjun Yu, Yabo Luo, Qingliang Yang, Yu Feng, Kaibin Yu, Jiaxue Li

**Affiliations:** 1State Key Laboratory of Public Big Data, Guizhou University, Guiyang 550025, China; gs.shuaipeng22@gzu.edu.cn (S.P.); gs.yufeng23@gzu.edu.cn (Y.F.); kbyu22@gzu.edu.cn (K.Y.); gs.jiaxueli22@gzu.edu.cn (J.L.); 2National Astronomical Observatories, Chinese Academy of Sciences (CAS), Beijing 10010l, China; bnsong@nao.cas.cn (B.S.); djyu@nao.cas.cn (D.Y.); yangqingliang@nao.cas.cn (Q.Y.); 3School of Traffic and Transportation Engineering, Changsha University of Science and Technology, Changsha 410114, China; luoyaborubber@163.com

**Keywords:** multi-sensor positioning, position prediction, LSTM-Self-Attention, GNSS/IMU, FAST feed cabin

## Abstract

The Five-hundred-meter Aperture Spherical Telescope (FAST), as the world’s most sensitive single-dish radio telescope, necessitates highly accurate positioning of its feed cabin to utilize its full observational potential. Traditional positioning methods that rely on GNSS and IMU, integrated with TS devices, but the GNSS and TS devices are vulnerable to other signal and environmental disruptions, which can significantly diminish position accuracy and even cause observation to stop. To address these challenges, this study introduces a novel time-series prediction model that integrates Long Short-Term Memory (LSTM) networks with a Self-Attention mechanism. This model can hold the precision of feed cabin positioning when the measure devices fail. Experimental results show that our LSTM-Self-Attention model achieves a Mean Absolute Error (MAE) of less than 10 mm and a Root Mean Square Error (RMSE) of approximately 12 mm, with the errors across different axes following a near-normal distribution. This performance meets the FAST measurement precision requirement of 15 mm, a standard derived from engineering practices where measurement accuracy is set at one-third of the control accuracy, which is around 48 mm (according to the accuracy form the official threshold analysis on the focus cabin of FAST). This result not only compensates for the shortcomings of traditional methods in consistently solving feed cabin positioning, but also demonstrates the model’s ability to handle complex time-series data under specific conditions, such as sensor failures, thus providing a reliable tool for the stable operation of highly sensitive astronomical observations.

## 1. Introduction

The Five-hundred-meter Aperture Spherical radio Telescope (FAST) is the world’s largest and most sensitive single-dish radio telescope, significantly advancing our ability to probe the universe’s deepest mysteries. With its excellent sensitivity, FAST detects faint signals from distant cosmic sources, substantially enhancing our understanding of the universe’s formation and evolution [1,2]. The feed cabin is an important part of the telescope. Accurate positioning of the feed cabin affects the telescope’s pointing accuracy. FAST’s feed cabin is different from others, and it is controlled by flexible steel ropes and always in motion. Currently, the measurement system of FAST’s feed cabin integrates the Global Navigation Satellite System (GNSS), Inertial Measurement Units (IMU), and Total Stations (TS). IMU is not affected by the external environment, but errors accumulate during IMU continuous measurement alone, so it cannot be used alone and must be used with other measuring devices, such as GNSS and TS, to maintain measurement accuracy. The feed cabin’s measurement system is enhanced by Kalman filtering techniques to provide high-precision positioning under optimal conditions.

However, these GNSS devices are vulnerable to orbital errors, satellite clock differences, and multipath effects, which can significantly degrade their accuracy [3,4]. Moreover, although Total Stations (TS) are employed for precision corrections, they are constrained by Earth’s curvature, atmospheric refraction, and their inherent equipment accuracies [5]. Maintaining such precision in diverse environmental conditions presents substantial challenges.

According to the Accuracy Analysis on Focus Cabin of FAST, the control accuracy of the feed cabin is approximately 48 mm [6]. Following the engineering practice where measurement precision is set to one-third of the control accuracy, the measurement precision requirement for FAST’s feed cabin has been set at 15 mm [7]. This standard is critical for ensuring the stable operation of the telescope, especially during long-term observations.

Advancements in filtering techniques have shown promise to maintain precision. Li et al. [8] introduced a combination of an extended Kalman filter and a particle filter. Additionally, Wu et al. [9] proposed a time-domain signal tracking and mitigation algorithm. Together, these advancements enhance positioning accuracy using UWB and foot-mounted IMUs, offering improvements over traditional methods. Furthermore, the integration of machine learning techniques has been explored to overcome the limitations of conventional systems. For instance, Chung et al. [10] and Fayyad et al. [11] have applied deep learning to better integrate data from multiple sensors for more accurate motion state recognition and positioning. Wang et al. [12] developed an enhanced deep learning network to fuse data from various sensors more effectively. Machine learning has the potential to surpass traditional methods by identifying complex patterns in data. However, deep learning approaches demand significant computational resources and large volumes of training data [13,14]. Additionally, the accuracy of these methods is contingent on the reliability of the devices; if the devices fail, the performance of the machine learning models can be compromised. In the practical application of the feed cabin measurement system, the GNSS receivers often fail because of interference from other satellite signals, and TS often also fail due to weather effects Wilgan et al. [15]. When these devices fail, even with the improved methods mentioned above, the measurement system must be stopped. To ensure telescope equipment safe, the astronomical observation mission must be shut down, too. This inevitably reduces the efficiency of the telescope. In order to prevent the measurement system from shutting down, this paper investigates a new method which predicts information about the position of the feed cabin after the devices failure based on the measurement data before the devices failure.

The development of Artificial Intelligence Generated Content (AIGC) technologies, including Transformer and Diffusion models, has further expanded the scope of machine learning applications [16]. In this process, time series forecasting research also involves various aspects, including agriculture, finance, industry, tourism, and so on [17,18,19,20]. Studies show that Spatiotemporal Convolutional Transformer networks can effectively manage complex time series data by leveraging the Self-Attention mechanism of Transformer models [21,22,23]. However, applying these advanced methods to the actual data and specific astronomical positioning requirements of the FAST telescope introduces challenges, such as high dimensionality and noise in astronomical data, drastic environmental fluctuations, and real-time processing issues [24]. To address these complexities and reduce model complexity, this paper proposes a novel time series prediction model that combines Long Short-Term Memory (LSTM) networks with a Self-Attention (SA) mechanism. LSTM is advantageous for prediction tasks because it can efficiently handle and remember long-term sequential data, making it suitable for processing complex data with time dependencies and excelling at capturing short-term and long-term dependencies [25,26]. Therefore, we employ it to predict the position information of the feed cabin in case of equipment failure. Nonetheless, LSTM has certain limitations, such as potential difficulties in capturing long-range dependencies and retaining information in long time series. Due to its reliance on recursive computation, LSTM is prone to issues like information decay and gradient vanishing when handling long time series [27,28,29]. The LSTM-Self-Attention method addresses these limitations by incorporating the Self-Attention mechanism, which enhances the model’s ability to capture global dependencies and more effectively retain and process crucial information in long time series [30,31,32]. Therefore, this paper combines LSTM with the Self-Attention method, improving the traditional model and enhancing its performance in handling long time series data. Using performance metrics such as Mean Absolute Error (MAE), Mean Squared Error (MSE), and R2, we compare this improved model with traditional BP and LSTM models. Results demonstrate that the LSTM-Self-Attention model not only significantly outperforms these methods, but also exceeds FAST’s current accuracy requirement of 15 millimeters.

This paper is organized as follows: Section 2 details the methodology, including data preprocessing (Section 2.1), model architecture (Section 2.2), training process (Section 2.4) and performance evaluation metrics (Section 2.5). Section 3 presents the results, and provides a comprehensive analysis of the model’s performance, which includes model performance (Section 3.1) and model comparison and analysis (Section 3.2). Particularly, in Section 3.2, we present a detailed analysis of the model’s performance metrics (Section 3.2.1), prediction results (Section 3.2.2), and errors (Section 3.2.3). Finally, Section 4 discusses the implications of our findings, and outlines potential areas for future research. This study aims to significantly enhance the precision and reliability of astronomical observations using the FAST telescope by addressing the challenges of environmental disturbances and the complex, nonlinear dependencies present in astronomical data.

## 2. Materials and Methods

This section systematically presents the complete methodology from data preparation to model validation. Initially, the FAST feed cabin dataset undergoes rigorous preprocessing to ensure accuracy and reliability. Subsequently, an advanced deep learning model, integrating LSTM and Self-Attention mechanisms, is designed and implemented, specifically targeting the characteristics of time-series data. Moreover, the training process of the model is elaborated, including optimization and adjustment strategies for parameters to ensure peak performance. Finally, the model’s predictive effectiveness in astronomical data analysis is comprehensively evaluated using multiple performance metrics, affirming the model’s practical applicability. The dataset, containing 129.6 million position data entries from the FAST feed cabin between January and May 2023, was obtained from the official FAST Observatory data archive, accessible at https://fast.bao.ac.cn.

### 2.1. Data Preprocessing

In this study, data preprocessing of the FAST feed cabin was conducted to ensure high accuracy and consistency in model training, validation, and testing; the entire process is detailed in Figure 1. The feed cabin’s movement state is complex and variable, encompassing 19 different modes of motion including tracking, source switching, and stationary combinations. This paper focuses on extracting precise data that reflect these primary motion states and appropriately processing them to meet the requirements of deep learning models. By analyzing the motion characteristics of the feed cabin, including Euclidean distance, displacements along three axes, velocity, and acceleration, we precisely extracted data representing tracking, source switching, and stationary states. This process involves filtering based on the duration of motion states using timestamps, particularly for the tracking state, with the following criteria established:The position must not be at the coordinate origin.The tracking speed must not exceed 20 mm/s, which is the maximum movement speed for this state.The duration of the tracking state must be at least one minute, corresponding to a minimum of 600 consecutive data points.Exclude data points with a velocity less than 1 mm/s to eliminate noise caused by factors such as wind vibration.

Initially, a check for missing values is performed and handled. For duplicate timestamps, the record with the smallest difference from neighboring data are retained and the other duplicates are removed. For missing data rows, the nearest neighbor method is employed for imputation. Spatial position charts and displacement and velocity graphs for the X, Y, and Z axes are plotted using Python 3.12.6 to identify and address anomalies. Subsequently, to enhance the model’s training efficiency and to meet the requirements of LSTM and Self-Attention models, every ten records are averaged to form a single record, thus consolidating 10 ms of data into a 1 s time series datum. Additionally, a moving average method as detailed in Equation (Equation 1) is utilized to construct a new time series format, where data from every 10 s is used to determine the characteristics of the subsequent second. For example, data from 1 to 10 s are averaged to form the data for the first second, data from 2 to 11 s are averaged for the second second, and so on. This supports the model’s learning of time dependencies. Finally, the processed dataset is divided into training, validation, and testing sets. The training set is used to train the model, the validation set for tuning model parameters and preventing overfitting, and the testing set to assess the model’s generalization ability on unseen data. This division strategy ensures that the test data are not exposed during the model’s training and validation phases, thereby accurately reflecting the model’s generalization capability and robustness.

### 2.2. Model Architecture

To accurately predict the position of the Five-hundred-meter Aperture Spherical Telescope (FAST) feed cabin, this study designed a deep learning model based on Long Short-Term Memory (LSTM) networks combined with a dual-head Self-Attention mechanism. This model architecture aims to fully leverage the sequential nature of time-series data, capturing long-term dependencies and interactions between important features within the sequence, while employing Self-Attention to focus on significant information in the data, enhancing the precision and efficiency of predictions. The model architecture diagram (see Figure 2) involves several machine learning frameworks.

#### 2.2.1. Long Short-Term Memory (LSTM)

The first part of the model consists of multiple LSTM layers, which handle the long-term dependencies in time series data. LSTM’s key feature is its ability to manage information flow through three gates: the Forget Gate (ft), Input Gate (it), and Output Gate (Ot), which work together to selectively retain or discard information as needed [33,34,35]. The model architecture diagram is shown in the Figure 3. This allows the model to capture both short- and long-term patterns in the data, avoiding the common issues of gradient vanishing or exploding faced by traditional RNNs [36]. In this study, each LSTM layer includes 256 hidden units across 8 layers, enabling the model to extract deep temporal features from the data. The input size for the LSTM layers is set to 3, corresponding to the X, Y, and Z coordinates of the FAST feed cabin. This architecture allows the model to learn from historical motion data of the feed cabin, which is critical for accurately predicting future positions in the event of sensor failure [37,38].
(1)SMAt=P1+P2+⋯+Pnn,n=10
(2)ft,i[C˜t]=σ(Wfi,c·[ht−1,x]+bfi,c)
(3)Ct=ft×Ct−1+it×C˜t
(4)ot=σ(Wo·[ht−1,x]+bo)
(5)ht=ot×tanh(Ct)

C˜t: Candidate Value Vector is created through a layer, and finally, combined with the forget and output gates, updates the information, as per Equation (Equation 3). Ot: Output Gate determines the output values for the next state. It assesses the current amount of information based on inputs from ht−1 and xt to decide what can be output. The current information Ct is processed through a layer, normalizing the values between −1 and 1, and is then multiplied by the output values Ot, to produce the final output ht, as per Equation (Equation 5).

#### 2.2.2. Self-Attention Mechanism

After processing the input through the LSTM layers, the model applies a dual-head Self-Attention mechanism to further refine the features extracted. Self-attention allows the model to focus on the most relevant portions of the time-series data by assigning attention scores to different time steps [39]. Using two attention “heads” in parallel enables the model to capture different aspects or relationships within the data, allowing it to allocate weights based on the importance of each time step [40], the model architecture diagram is shown in the Figure 4. The Self-Attention mechanism calculates Query, Key, and Value vectors for the input sequence and computes attention weights, which are then used to generate the final output through a weighted summation.
(6)Attention(Q,K,V)=softmaxQKTdkV
where Q=XWQ,K=XWK,V=XWV are the linear transformations of the queries, keys, and values, WQ,WK,WV is a learnable parameter matrix, and dt is a scaling factor to control the magnitude of the inner products, dk typically the square root of the dimension of the keys. In the dual-head Self-Attention mechanism, two sets of Q, K, V are computed in parallel, corresponding to two different attention “heads”. Each head captures different aspects or feature combinations of the input data. The output of the dual-head Self-Attention is either the concatenation or another form of combination of the outputs from the two heads. Thus, for dual-head Self-Attention, it can be expressed as
(7)MultiHead(X)=Concat(head1,head2)WO
(8)headi=Attention(XWiQ,XWiK,XWiV)
where WiQ,WiK,WiV are the ith parameter matrix of the head, where WO is a parameter matrix combining the outputs from different heads. Each head follows the standard Self-Attention formula, but can have its own independent parameters, allowing them to learn different feature representations of the input data. By combining these different representations, the dual-head Self-Attention mechanism provides a richer, more complex capability for processing information, merging the outputs of the two heads to achieve a more enriched feature representation of the sequence. This computational approach enables the model to automatically highlight the most important temporal data features in the current observation tasks of the FAST feed cabin, enhancing prediction accuracy.

In the dual-head Self-Attention mechanism, two sets of Query, Key, and Value matrices are computed simultaneously, allowing each head to capture distinct aspects of the data. The results from the two attention heads are then concatenated to form a more comprehensive representation of the sequence. This mechanism is particularly effective for tracking irregular, nonlinear patterns in the FAST feed cabin data, improving the model’s predictive performance. By highlighting the most important temporal data features, the dual-head Self-Attention mechanism enhances the accuracy of the feed cabin’s position predictions.

#### 2.2.3. LSTM-Self-Attention

In this study, we employed Long Short-Term Memory (LSTM) due to its ability to handle long-term dependencies in sequential data, a crucial factor in accurately predicting the FAST feed cabin’s movement over time. LSTM’s gated mechanisms (Input Gate, Forget Gate, and Output Gate) allow it to overcome issues of gradient vanishing or exploding, which traditional RNNs struggle with when processing long sequences. LSTM was chosen specifically because of its strength in retaining important information from earlier time steps, which makes it ideal for tasks like position prediction where long-term relationships are key [41]. However, LSTM alone has limitations in capturing global dependencies within sequences, which is why we integrated it with a Self-Attention mechanism, forming the LSTM-Self-Attention (LSTM-SA) model. The Self-Attention mechanism enhances the model’s ability to selectively focus on important parts of the input sequence, which is critical in predicting the feed cabin’s position when dealing with complex time-series data. This hybrid approach allows the model to maintain high accuracy, even when some sensors fail. The LSTM-SA model construction involves the following steps:The LSTM layers first process the input time-series data (such as historical position and velocity information), capturing both short- and long-term dependencies. The LSTM layers are used to process the sequential data, which may include the feed cabin’s past position and velocity measurements. LSTM is designed to retain long-term dependencies through its three gates (Input, Forget, and Output Gates), ensuring that relevant information from earlier time steps is used to make predictions at later points.After passing through the LSTM layers, the data are fed into the Self-Attention mechanism. The Self-Attention mechanism assigns different weights to different time steps, focusing on the most relevant inputs for prediction. This allows the model to ‘attend’ to critical features in the sequence. The Self-Attention mechanism works by assigning attention scores to each time step in the sequence, allowing the model to focus on the most important features of the input sequence, such as critical position changes. This enables the model to give higher importance to relevant time steps and less importance to redundant or irrelevant data.To enhance the model’s generalization, we use a dual-head attention mechanism. This means that the attention mechanism evaluates the input sequence from two different perspectives, or ‘heads’. Each head focuses on different aspects of the sequence, improving the model’s ability to capture complex dependencies over time.Finally, the output from the Self-Attention layers is passed through a fully connected layer to predict the remaining trajectory and position of the feed cabin, which converts the weighted sequence data into the final prediction of the feed cabin’s future position.

In summary, the LSTM-Self-Attention (LSTM-SA) model combines LSTM’s strength in processing sequential data with the Self-Attention mechanism’s ability to highlight important features in long time-series data. This combination significantly improves the model’s predictive accuracy, especially under sensor failure conditions.

#### 2.2.4. Model Fusion and Output

After processing by the dual-head Self-Attention mechanism, the feature representations are fed into the model’s fusion layer—the Fully Connected (FC) layer. The task of the FC layer is to further synthesize and refine the high-level features extracted by the LSTM and Self-Attention layers through a series of linear transformations and nonlinear activation functions, outputting precise predictions for the future position of the feed cabin as per Equation (Equation 9). The output size of the FC is also set to 3, corresponding to the predicted X, Y, and Z coordinates of the FAST feed cabin. Finally, this study employs the Mean Squared Error (MSE) as the loss function to quantify the difference between the model’s predictions and the actual values as per Equation (Equation 10).
(9)f(x)=∑i=1dwixi+b
(10)MSE=1n∑i=1n(f(xi)−yi)2

The optimizer chosen is the Adam optimizer, whose adaptive learning rate adjustment effectively accelerates the training process and enhances the model’s convergence speed. A learning rate scheduler is also added during training to automatically adjust the learning rate based on the loss on the validation set, further enhancing training efficiency and model performance. Additionally, an early stopping strategy is implemented to prevent overfitting, ensuring the model’s generalization ability during training.

### 2.3. Handling Device Failures with the LSTM-SA Model Mechanisms

In the practical application of the FAST feed cabin measurement system, GNSS receivers frequently fail due to overlapping satellite signal interference, while Total Stations (TS) are susceptible to environmental factors like rain, fog, and snow, which degrade performance. These weather conditions obstruct optical signal transmission, reducing the accuracy of TS measurements. The LSTM-SA model addresses these issues by using historical measurement data from before the device failure to predict the feed cabin’s position after the failure. This allows the model to continuously provide reliable position data, avoiding system shutdown.

The movement of the FAST feed cabin can be categorized into twelve types from tables as, e.g., Table 1 shows, with each associated with a known trajectory. Based on the data collected before the device failure, the system can identify the current movement mode of the feed cabin. The LSTM-SA model then uses this historical data, including spatial position and velocity information, as input to predict the remaining trajectory and future position of the feed cabin. Even during GNSS or TS device failures, this prediction ensures that the measurement system can continue providing accurate position data to the telescope. When GNSS or TS device failures are detected—identified through anomaly detection methods that spot sudden deviations from the expected smooth trajectory—the system automatically switches to a prediction-based mode using the LSTM-SA model. At this point, erroneous data from the faulty devices is ignored, and the predicted position data are transmitted to the telescope’s control system to maintain operational stability and continuity.

### 2.4. Training Process

In this study, we detail the training process of a deep learning model designed to precisely predict the position of the Five-hundred-meter Aperture Spherical Radio Telescope (FAST) feed cabin. This model integrates Long Short-Term Memory networks (LSTM) with a Self-Attention mechanism, aimed at handling and learning complex time series data. Our dataset originates from 129.6 million position data entries collected from the feed cabin from January to May 2023. Before training the model, we loaded and processed five CSV format files (preprocessed data from the FAST feed cabin from January to May), including standardization and construction of time series data, ensuring that features at different scales were effectively learned, and allowing the model to capture the temporal dependencies in the data. Through a custom-designed CustomDataset class, the data were further processed, including serialization and division into training, validation, and test sets. Our model architecture includes an 8-layer LSTM network with 256 hidden units, supplemented by a dual-head Self-Attention mechanism, with predictions outputted through fully connected layers. We employed two Self-Attention mechanisms to capture key points in the time series data, using Mean Squared Error (MSE) as the loss function. The model was trained using the AdamW optimizer, with an initial learning rate set at 0.0001 and batch size adjusted to 64. Coupled with the ReduceLROnPlateau learning rate scheduler that dynamically adjusts the learning rate based on validation loss, and introducing an early stopping mechanism to prevent overfitting, we maintained the model’s generalization ability.

During the training process, the model underwent iterative training on a training set with a batch size of 64, continuing for 5 major cycles and 300 minor cycles, totaling 1500 training epochs, or until the early stopping conditions were met. We assessed model performance through the validation set and adjusted the learning rate based on validation loss. To more visually assess the training process and monitoring strategies, we used the Matplotlib library to plot charts showing the changes in loss and learning rate over the training cycles, demonstrating the model’s training efficiency and convergence, for more intuitive parameters see Table 2.

### 2.5. Performance Evaluation Metrics

To comprehensively assess the accuracy and effectiveness of the LSTM-Self-Attention model in predicting the position of the Five-hundred-meter Aperture Spherical Radio Telescope (FAST) feed cabin, we employed three common performance evaluation metrics: Root Mean Squared Error (RMSE), Mean Absolute Error (MAE), and R2 score (R-squared). These metrics provide us with a clear framework for quantifying the quality of the model’s predictions, allowing us to thoroughly analyze the model’s performance.

Root Mean Squared Error (RMSE): RMSE is a standard metric for assessing the difference between model predictions and actual values. It calculates the average of the squared differences between the predicted errors, reflecting the accuracy of the model’s predictions. The lower the RMSE, the closer the model’s predictions are to the actual values, indicating better predictive performance. Its calculation formula is
(11)RMSE=1n∑i=1n(yi−y^i)2

Mean Absolute Error (MAE): MAE measures the average of the absolute differences between the predicted and actual values, providing an intuitive measure of predictive accuracy. Compared to RMSE, MAE is less sensitive to outliers, offering another perspective on the model’s predictive performance. Its calculation formula is
(12)MAE=1n∑i=1n|yi−y^i|

R2 (R-squared) measures how much of the variability in actual values can be explained by the variability in model predictions. The closer the R2 value is to 1, the higher the model’s explanatory variability, indicating better predictive performance. It is an important metric for evaluating model fit, especially suitable for regression problems. Its calculation formula is
(13)R2=1−∑i=1n(yi−y^i)2∑i=1n(yi−y¯)2
where *n* is the total number of samples, yi is the actual value, y^i is the predicted value, and y¯i is the mean of yi.

These three metrics together form the basis for our evaluation of the LSTM-Self-Attention model’s performance in the task of time series prediction. By integrating the results from these metrics, we can gain a comprehensive understanding of the model’s accuracy, stability, and robustness in predicting the position of the FAST feed cabin. These evaluation results not only demonstrate the effectiveness of the model architecture, but also provide important references for future improvements to the model and its application to similar problems.

## 3. Results

In this section, we will discuss in detail the performance of three models (BP, LSTM, LSTM-SA) in predicting the position of the FAST feed cabin. Initially, we will present the overall prediction results of these models on the complete dataset, followed by an in-depth analysis of model errors and exploration of key factors affecting prediction performance. Through this comprehensive evaluation, we aim to provide robust data support for optimizing model structures and parameter selection.

### 3.1. Model Performance

In this study, we introduced an LSTM-Self-Attention model that combines Long Short-Term Memory networks and Self-Attention mechanisms aimed at accurately predicting the feed cabin position of the Five-hundred-meter Aperture Spherical Telescope (FAST). The model was trained and validated using approximately 129.6 million data points, with a time interval of 0.1 s between each point, collected from January to May 2023.

In this model training process, the performance of indicators such as Loss, RMSE, MAE, and R-squared revealed the model’s convergence and optimization. As shown in Figure 5, during the first 100 epochs of training, the model’s Loss significantly decreased to 2.65 ×10−6, RMSE decreased from 2.92 ×10−3 m to 0.47 ×10−3 m, and R-squared gradually increased, indicating that the model was effectively learning and adjusting its parameters to fit the training data. During this phase, the model quickly transitioned from a random initialization state to progressively learning the data features and reducing prediction errors. Between epochs 100 and 110, the Loss experienced the greatest decrease, dropping from 2.65 ×10−6 to 1.75 ×10−6. RMSE also decreased rapidly, and R-squared surged from 0.9143 to 0.9847. This suggests that the model achieved a significant learning breakthrough during this phase, possibly resolving a key challenge in the training data and making crucial weight adjustments that substantially improved the model’s fit. From epochs 110 to 160, Loss and RMSE continued to decline, albeit at a slower rate. This gradual deceleration suggests that the model was nearing its optimal state as the extraction of effective features diminished with further optimization. The Loss stabilized around 9.77 ×10−7, while RMSE and MAE also approached steady levels. After more than 250 epochs, the decrease in Loss was extremely slow, eventually converging at 2.23 ×10−7. This indicates that the model had essentially converged, having found a reasonably good global or local optimum, with further training yielding negligible improvements in Loss. The mean error was 1.58 mm, well below the target of 15 mm. Although MAE fluctuated during training, occurring at rounds 101, 104, 131, 148, 162, 255, and 268, it ultimately remained at a low error level. R-squared steadily increased throughout the process, ultimately reaching 0.997, indicating that the model had a high explanatory power of variance and predictive accuracy. Moreover, the adjustments in optimization strategies and changes in data feature complexity at critical stages enabled the model to leap from local optima to more optimal regions. After more than 250 epochs of training, the model achieved the desired optimization, realizing the objectives of high accuracy and stability in predictions.

These results demonstrate that the LSTM-Self-Attention model demonstrated effective performance in predicting the feed cabin position of the FAST telescope and is an effective tool for such time-series prediction problems. Future work may include further feature engineering, hyperparameter tuning, or the use of more complex model architectures to enhance model performance.

### 3.2. Comparison and Analysis

To comprehensively evaluate the performance of our proposed LSTM-Self-Attention (LSTM-SA) model, we compared it against conventional time series prediction models, including the Backpropagation Neural Network (BP) and the standard LSTM model. By analyzing the performance metrics and prediction results on the same dataset, we aimed to demonstrate the advantages of the LSTM-SA model in predicting the position of the FAST telescope’s feed cabin.

#### 3.2.1. Comparison of Models’ Performance Metrics

In assessing model performance, the most commonly used metrics include Mean Squared Error (MSE), Root Mean Squared Error (RMSE), Mean Absolute Error (MAE), and the coefficient of determination (R2). These metrics quantify the difference between predicted and actual values and are essential tools for assessing model accuracy. To facilitate a more intuitive comparison of model performance, this study utilized data from January to May 2023 (comprising 13,046,400 data points at 1 s intervals) for training and validation. The four metrics—MSE, RMSE, MAE, and R2—were evaluated, and radar charts were generated based on these scores, as illustrated in Figure 6.

For a comprehensive evaluation of the BP, LSTM, and LSTM-SA models, during the calculation of the values for each metric in the radar chart, the R2 value was kept unchanged, while the MSE, RMSE, and MAE values were adjusted by subtracting each from 1; the closer to 1, the better the result. The final goal was to normalize all metrics within the range of 0.9 to 1, making the visualization more intuitive. According to the results, the LSTM model scored 0.999, 0.971, and 0.981 on the MSE, RMSE, and MAE metrics, respectively. The LSTM-Self-Attention (LSTM-SA) model scored 0.99974, 0.984, and 0.990 on the same metrics. In contrast, the BP model scored 0.996, 0.936, and 0.964. These results indicate that the LSTM-SA model slightly outperforms the other models in the key error metrics of MSE, RMSE, and MAE, particularly in RMSE and MAE. This suggests that the LSTM-SA model has a superior ability to minimize errors, leading to smaller discrepancies between predicted and actual values. In terms of the coefficient of determination (R2), the LSTM, LSTM-SA, and BP models achieved scores of 0.992, 0.996, and 0.995, respectively. Although these values are all very close to 1, indicating that all models can accurately capture data variations, the LSTM-SA model’s slightly higher R2 score further validates its superior accuracy. Overall, the LSTM-SA model performs the best across all error metrics (MSE, RMSE, and MAE), exhibiting the smallest errors. The LSTM model follows, with the BP model performing relatively poorly. The LSTM-SA model’s superior performance in these metrics may be attributed to the introduction of the Self-Attention mechanism, which enhances the model’s ability to capture long-range dependencies in sequential data, thereby improving prediction accuracy. Additionally, while the differences in R2 are minimal, the LSTM-SA model still exhibits slightly better performance. This indicates that the LSTM-SA model not only excels in minimizing errors, but also offers greater reliability in overall prediction capability and model interpretability.

In summary, based on the analysis of these metrics, the LSTM-SA model demonstrates outstanding performance and emerges as the optimal predictive model in the current study, particularly suitable for applications requiring high-precision forecasting.

#### 3.2.2. Comparison of Model’ Prediction Results

By predicting the June 2023 data (comprising 2,592,000 data points with a time interval of 1 s between each point), the predicted results were compared with the actual values from June, and the RMSE and MAE variation over time were plotted. The mean values of these metrics were then calculated to assess the predictive performance of the BP, LSTM, and LSTM-SA models, as shown in Figure 7.

The figure clearly shows that the difference between the predicted value and the real value of BP model’s consistently exceeded the target value of 15 mm, with a maximum prediction of 196 mm, indicating the poorest performance and failing to meet practical usage requirements. Although the LSTM model performed better, with most predictions below 25 mm, it still did not meet the 15 mm target, and the maximum error reached 106 mm. In contrast, the LSTM-SA model demonstrated superior performance, with the majority of predictions below 15 mm and a maximum error of 85 mm. Overall, compared to the BP model’s average error of 38 mm, the LSTM and LSTM-SA models reduced the average error by 58% and 74%, respectively. While the LSTM and LSTM-SA models exhibited similar prediction trends, the introduction of the Self-Attention mechanism in LSTM-SA led to a significant reduction in the maximum error from 106 mm to 85 mm and a 37.5% decrease in the average error from 16 mm to 10 mm, meeting the practical requirement of an error below 15 mm. Furthermore, the MAE plot shows that the introduction of the Self-Attention mechanism makes the prediction results of LSTM-SA more stable, with its maximum MAE significantly lower than LSTM’s 0.289 and BP’s 0.295. The average MAE values are 0.006, 0.012, and 0.023, representing improvements of 50% and 74% for LSTM-SA compared to LSTM and BP, respectively.

Overall, in the June data used for prediction, significant volatility was observed from 1 to 5 June and from 25 to 30 June, and the model’s fit during these periods was suboptimal. As shown in Figure 7, these volatile data points are concentrated in the regions corresponding to X coordinates between 0 and 500,000 and beyond 2,500,000. This phenomenon reflects the complexity of the feed cabin data, indicating that the model tends to capture the more intricate data features during the training process.

During the beginning and end of the month, maintenance and repairs for the feed cabin typically shorten the observation durations, resulting in fewer data features and less informative predictions. This produces relatively simple data, which has minimal impact on model training. This further suggests that the model’s training primarily relies on complex data, while simple data contributes minimally to improving model performance. This finding is crucial for optimizing model training strategies, highlighting the need to focus more on effectively handling complex data during observation tasks. At the beginning and end of the month, when maintenance and repairs typically occur, the shorter observation durations result in fewer data features, leading to less informative predictions. However, during the middle of the month, particularly within the 1 to 2 million data points range, even the least effective BP model showed improved fitting results, while LSTM and LSTM-SA models became more stable. Notably, between 2.1 and 2.4 million data points, the errors for BP and LSTM-SA models were nearly zero, with BP performing well than LSTM-SA under stable or tracking motion conditions, further exemplifying the higher performance of BP models in datasets with less data volatility. Despite the diversity of FAST observation tasks, LSTM and LSTM-SA models consistently outperformed the BP model, offering more stable predictions. When comparing LSTM with LSTM-SA specifically, the Self-Attention mechanism further enhanced prediction accuracy, particularly towards the end of the month when the number of observation tasks and motion patterns decreased. The 15 mm standard line in the figure illustrates that the LSTM-SA model significantly improved the predictive precision over the LSTM model, highlighting the advantages of the Self-Attention mechanism.

In addition to evaluating the model’s performance using MSE, MAE, R2, and RMSE, we have visualized the time series prediction results for the FAST feed cabin under different models. Figure 8 illustrates the predicted vs. actual positions for the LSTM, LSTM-SA, and BP models, providing a clear comparison of the models’ predictive accuracy over time. The LSTM-SA model outperforms the LSTM and BP models in terms of both prediction accuracy and error distribution, as evidenced by the lower MSE values and more accurate trajectory predictions. The incorporation of the Self-Attention mechanism in the LSTM-SA model allows it to capture global dependencies within the time series, leading to more accurate position predictions for the FAST feed cabin.

In summary, the LSTM-SA model demonstrates the best performance in predicting June 2023 data compared to BP and LSTM models. It achieves a substantial reduction in both maximum and average errors, meeting the practical accuracy requirements. The Self-Attention mechanism in LSTM-SA enhances prediction stability and accuracy, particularly at the end of the month, and shows a significant improvement over the LSTM and BP models.

#### 3.2.3. Error Analysis

In practical applications, the analysis of prediction errors is crucial for achieving real-time correction and validation of the predicted results. Through the performance analysis of the aforementioned models and understanding of prediction accuracy, we found that although the LSTM-SA model’s overall error accuracy meets the target value of 15 mm, the overall performance of the three models is suboptimal in scenarios with fewer observational tasks and simpler motion patterns, such as at the beginning and end of the month. Therefore, error analysis for data prediction in practical applications should be based on the range where the model predictions are stable, rather than merely analyzing overall error. Furthermore, for scenarios where the positional accuracy of the FAST feed cabin is critical, it is necessary to analyze the impact of errors in detail along the X, Y, and Z axes, providing a basis for subsequent research on auxiliary correction of measurement errors. To this end, in the error analysis, we excluded the data from the first and last five days of June, and used the three models to predict the data along the X, Y, and Z axes for 20 days in the middle of the month, sampling one prediction every 5 h to calculate the prediction error. The final error analysis plots of the three models BP, LSTM and LSTM on X, Y and Z axes are obtained, as shown in Figure 9. This approach not only allows for better analysis of the model’s predictive performance within the stable error range but also provides a comprehensive assessment of the impact of each axis on the overall error, offering scientific guidance for model selection and optimization in practical applications.

Based on the analysis of the error distribution plots, during continuous iterative training, all three models exhibit a normal distribution in their results. Specifically, the BP model’s error distribution, across the X, Y, and Z axes, is concentrated within 3 mm. In contrast, the LSTM model shows higher error density peaks, with the X-axis at 10 mm, the Y-axis at 34 mm, and the Z-axis at 18 mm, indicating a less favorable overall error density distribution. However, by introducing the Self-Attention mechanism in the LSTM-SA model, the issue of error center axis deviation seen in the LSTM model is effectively mitigated, bringing the error distribution back within 3 mm, demonstrating the advantage of the Self-Attention mechanism in refining error density distribution. In addition, in terms of error stability, the BP model performs significantly worse than both the LSTM and LSTM-SA models, with a standard deviation ranging between 23.9 mm and 83.1 mm, compared to 9.4 mm to 30.5 mm for LSTM, and 3.9 mm to 8.6 mm for LSTM-SA, indicating that the latter two models offer superior stability in error distribution. Based on the analysis of the 95% confidence intervals for the BP, LSTM, and LSTM-SA models across the X, Y, and Z axes, several key observations can be made. For the BP model, the 95% confidence intervals are [−0.1138, 0.1138] for the X-axis, [−0.1367, 0.1367] for the Y-axis, and [−0.0393, 0.0393] for the Z-axis. These relatively wide intervals suggest a higher degree of variability and lower prediction accuracy, particularly along the Y-axis, indicating that the BP model struggles to maintain consistency across different axes. In contrast, the LSTM model demonstrates narrower confidence intervals, with [−0.0265, 0.0265] for the X-axis, [−0.0502, 0.0502] for the Y-axis, and [−0.0155, 0.0155] for the Z-axis. These results reflect improved accuracy and stability compared to the BP model, particularly along the X and Z axes, though the Y-axis still exhibits greater variability, which may be attributed to the inherent complexity of the data along this axis. The LSTM-SA model further refines these results, achieving the narrowest confidence intervals: [−0.0128, 0.0128] for the X-axis, [−0.0141, 0.0141] for the Y-axis, and [−0.0064, 0.0064] for the Z-axis. This significant reduction in confidence interval width indicates superior prediction accuracy and consistency across all axes, with the Z-axis showing the least variability, likely due to the more stable motion pattern along this axis.

Analyzing the axes separately, the overall error for the Z-axis is smaller than that for the X and Y axes. This is likely due to the better stability of the Z-axis during the movement of the FAST feed cabin, where it primarily moves vertically with fewer disturbances and less variability, making it more suitable for time series analysis. Consequently, the LSTM model performs better than the BP model for this type of data. The introduction of the Self-Attention mechanism further concentrates and evens out the LSTM model’s error distribution, thereby reducing errors. Conversely, the LSTM model’s predictions for the X and Y axes are less accurate than those of the BP model, suggesting that noise and outliers in the original data adversely affect the model’s performance, despite data cleaning and normalization. The LSTM model is more sensitive to these anomalies, which reduces prediction accuracy, whereas the BP model is less affected. When the Self-Attention mechanism is added to the LSTM model, the resulting LSTM-SA model not only integrates LSTM’s long-term memory capabilities and enhanced feature capture but also improves prediction accuracy by reducing the impact of noise and outliers. However, adjustments to parameters such as the number of hidden layers and learning rate, particularly the number of heads in the Self-Attention mechanism, are still required to optimize performance, as shown in Figure 10.

Finally, in comparing the three models, the LSTM-SA model (represented in red) demonstrates the highest alignment with the actual data, outperforming both the BP (black) and LSTM (gray) models, as shown in Figure 9. The LSTM-SA model also exhibits smaller confidence intervals across all axes, underscoring its ability to minimize errors and provide more reliable predictions. Moreover, the LSTM-SA model more accurately captures the positional changes and movement patterns of the FAST feed cabin. These findings highlight the effectiveness of incorporating the Self-Attention mechanism into the LSTM model, particularly in reducing the impact of noise and outliers. In contrast, the BP neural network and standard LSTM models often exhibit deviations and delays in their prediction curves, leading to inaccuracies. The introduction of the Self-Attention mechanism allows the LSTM-SA model to better handle changes in observational tasks and maintain superior overall performance. The results suggest that the LSTM-SA model is particularly well-suited for applications requiring high-precision predictions, especially when the data exhibit complex dependencies and varying levels of noise. The LSTM-SA model’s ability to quickly respond to these changes ensures that it maintains both accuracy and timeliness in its predictions.

## 4. Discussion and Conclusions

This study presents a novel approach for predicting the position of the FAST feed cabin by integrating Long Short-Term Memory (LSTM) networks with a Self-Attention (SA) mechanism, aiming to overcome the limitations of traditional GNSS and TS-based positioning systems that suffer under adverse environmental conditions. Our research demonstrates that the proposed LSTM-SA model significantly outperforms conventional methods, including Backpropagation (BP) and standard LSTM models, in terms of accuracy, stability, and robustness.

### 4.1. Key Findings

#### 4.1.1. Model Performance

The LSTM-SA model exhibited superior performance across all evaluated metrics, including Mean Absolute Error (MAE), Root Mean Square Error (RMSE), and R2. Specifically, the model achieved an MAE of less than 10 mm and an RMSE of approximately 12 mm, comfortably meeting the FAST operational precision requirement of 15 mm. In contrast, the BP and LSTM models produced larger errors, indicating their lesser suitability for this application.The LSTM-SA model’s enhanced capability to capture long-range dependencies and reduce the impact of noise and outliers, thanks to the Self-Attention mechanism, is a key factor in its superior performance.The LSTM-SA model demonstrates strong predictive capabilities for the FAST feed cabin time-series data, particularly in scenarios involving sensor failures. By integrating long-term dependency capture with the LSTM network and the global feature extraction capability of the Self-Attention mechanism, the model achieves high prediction accuracy. However, the LSTM-SA model is specifically designed for the FAST feed cabin system, and may not be robust to changes in internal structure or external environments beyond this context. Future work could explore the model’s adaptability to different systems or external variables to improve its overall robustness.

#### 4.1.2. Error Analysis

Detailed error analysis revealed that The overall mean of the error distribution of the LSTM-SA model in the X, Y, and Z axes is concentrated about 3 mm, 95% of the errors are less than 14.1 mm, as can be seen in Figure 9, demonstrating greater precision and stability compared to the BP and LSTM models. The model’s standard deviation in error was notably lower, indicating its robustness in varying conditions.The model’s performance was particularly strong in predicting the Z-axis position, which is generally more stable during the feed cabin’s motion. This suggests that the model can effectively handle less complex motion patterns, making it suitable for diverse observational tasks.

### 4.2. Implications and Future Work

The findings of this study have significant implications for the operational efficiency of the FAST telescope. By ensuring high-precision predictions even when traditional measurement devices fail, the LSTM-SA model can help prevent disruptions in astronomical observations, thereby maximizing the telescope’s usage and efficiency. The model’s robustness also makes it a promising tool for other applications requiring accurate time-series predictions under challenging conditions. For future research, several avenues can be explored to further enhance the model’s performance:The analysis from tables as, e.g., Table 1 shows that the model’s prediction accuracy is closely related to the motion modes of the feed cabin. The variation in correlation across different motion modes indicates that the model performs well in capturing position trends in some modes, such as MultiBeamOTF and OnTheFlyMapping, but poorly in others, like Tracking and SwiftCalibration. Observation time does not directly correlate with model performance, as seen in the Tracking mode, where despite the longest observation duration, the correlation is low. Future work will focus on improving the model architecture, designing specialized models for specific modes, balancing the dataset, and further analyzing sources of error to enhance overall prediction performance and generalization.Addressing model complexity and data dependency: The dual-head Self-Attention mechanism is computationally demanding and reliant on large volumes of high-quality data. These challenges could be mitigated by adopting more efficient architectures and enhanced data augmentation strategies.Enhancing feature engineering and real-time implementation: Incorporating additional factors like environmental conditions and more complex motion patterns could further boost the model’s predictive accuracy. Implementing the model in a real-time system for feed cabin positioning could help evaluate its practical applicability and identify areas for further refinement.

The LSTM-Self-Attention model, in addressing the challenges of feed cabin positioning for the FAST telescope, offers a reliable and precise solution. By exceeding current operational standards, and providing a robust, reliable, and efficient predictive tool, this model underscores the potential of integrating advanced machine learning techniques with traditional sensing systems to address complex scientific instrumentation challenges. Future enhancements focusing on model adaptability and efficiency could broaden its application scope and impact across various scientific and technological domains.

## Figures and Tables

**Figure 1 sensors-24-06857-f001:**
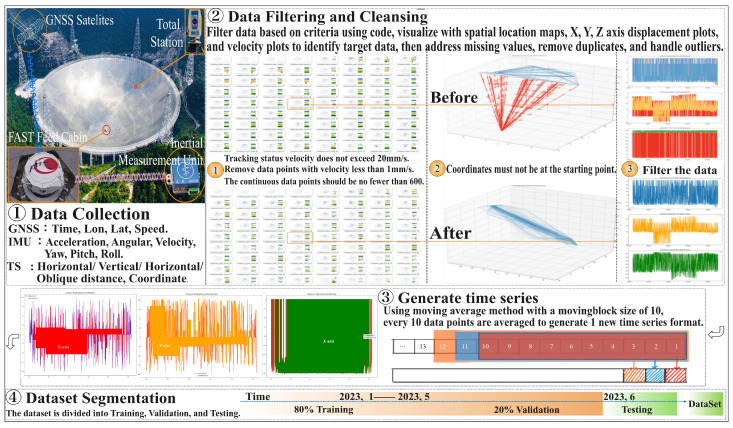
FAST feed cabin data preprocessing: this illustrates the entire process of data collection, cleaning, processing, and analysis using the FAST telescope, encompassing four main stages: data collection, data filtering and cleaning, data integration, and dataset segmentation.

**Figure 2 sensors-24-06857-f002:**
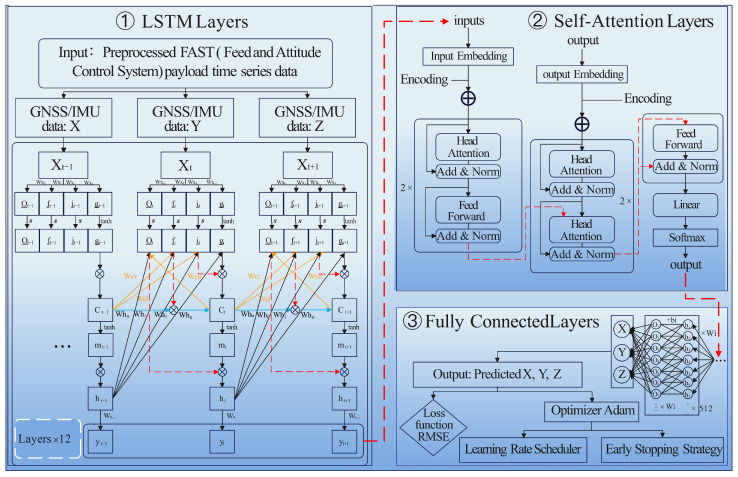
Illustration of the architecture of the machine learning component of this study, divided into three main parts: LSTM Layers are designed to handle and predict long-term dependencies in time-series data; Self-Attention Layers utilize a multi-head attention mechanism to improve gradient flow during training and accelerate convergence; Fully Connected Layers employ Mean Squared Error (MSE) as the loss function to optimize model parameters and integrate the Adam optimizer with a learning rate scheduler, as well as an early stopping strategy to prevent overfitting and enhance model performance.

**Figure 3 sensors-24-06857-f003:**
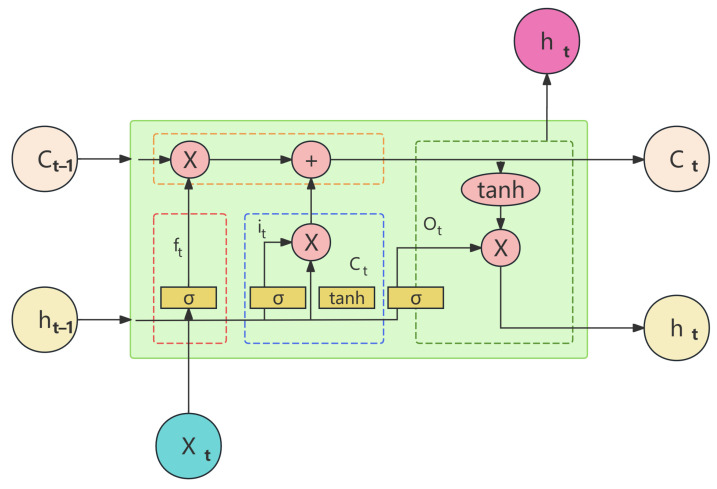
Schematic of LSTM model architecture.

**Figure 4 sensors-24-06857-f004:**
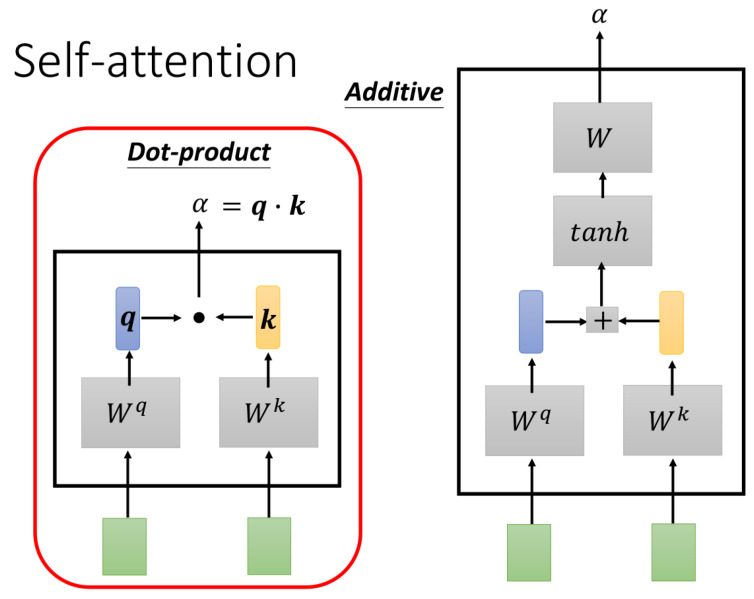
Schematic of Self-Attention model architecture.

**Figure 5 sensors-24-06857-f005:**
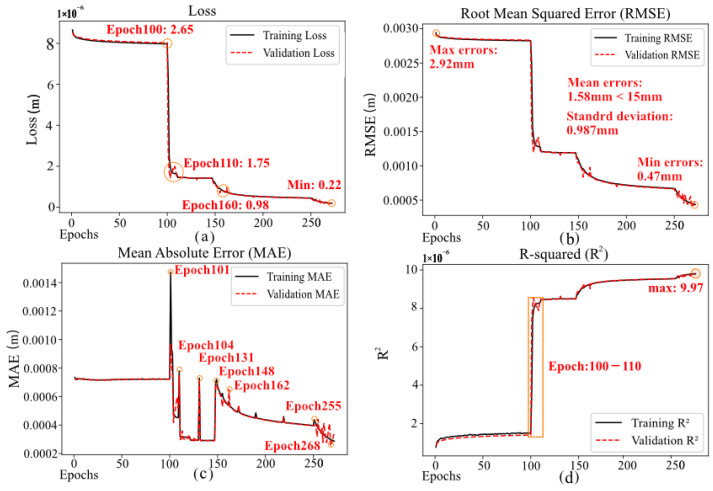
The image illustrates the performance metrics of the machine learning model across various training epochs. Each graph contains two lines, with the solid line showing results on training data and the dashed line on validation data. (**a**) This graph depicts the model’s loss values at different training epochs, illustrating how the model’s errors decrease with continued training. (**b**) This graph presents the Root Mean Square Error (RMSE) for training and validation, which measures the discrepancies between actual and predicted values. (**c**) The graph shows the Mean Absolute Error (MAE) for training and validation, reflecting the average level of prediction errors. (**d**) The final graph displays the coefficient of determination (R2), which assesses the model’s ability to explain the variability in the data.

**Figure 6 sensors-24-06857-f006:**
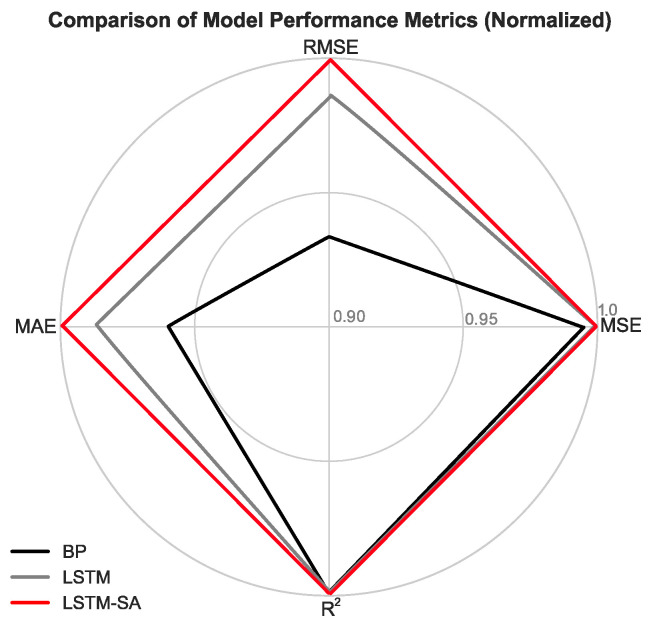
The radar chart illustrates the overall performance of three models (BP, LSTM, LSTM-SA) in time series forecasting, using metrics such as standardized Mean Squared Error (MSE), Root Mean Squared Error (RMSE), Mean Absolute Error (MAE), and R-squared (R2). For example, if the LSTM model’s MSE, RMSE, and MAE are 0.00085, 0.02923, and 0.19508 m, respectively, the normalized scores for these metrics would be calculated as 1 − 0.00085, 1 − 0.02923, and 1 − 0.19508, resulting in scores of 0.99915, 0.97077, and 0.80492. The R2 value remains unchanged. The final goal was to normalize all metrics to fall within the range of 0.9 to 1, making the visualization more intuitive.

**Figure 7 sensors-24-06857-f007:**
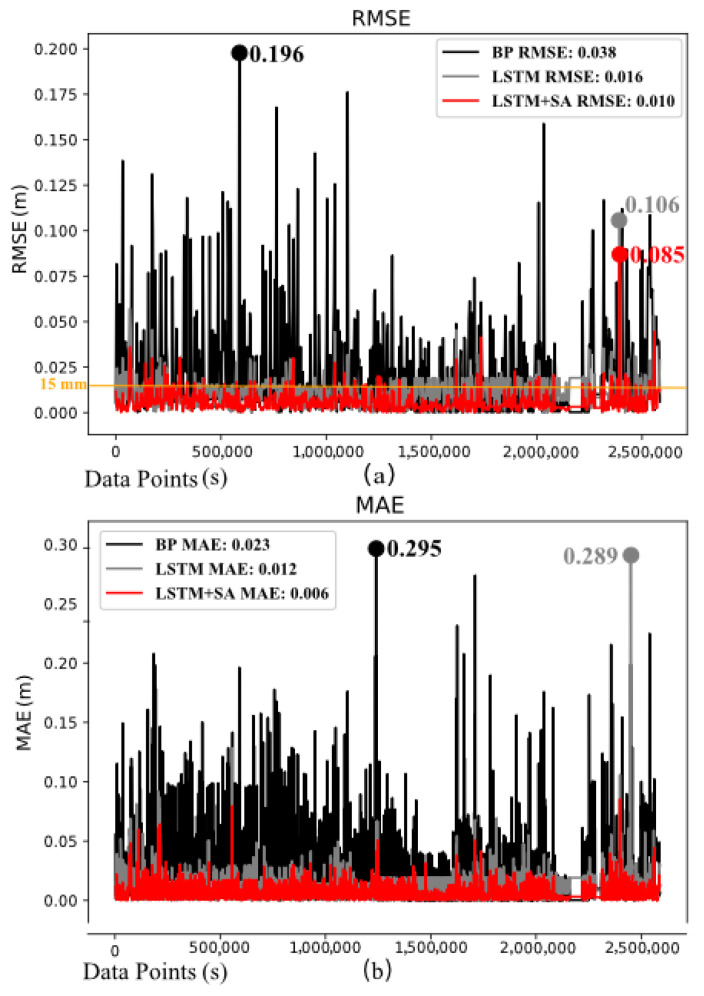
The images illustrate the results of BP, LSTM, and LSTM-SA models trained on a dataset from January to May 2023, and then used to predict the results for the June 2023 dataset. Each image contains three types of lines: black for BP, gray for LSTM, and red for LSTM-SA. (**a**) shows the RMSE prediction results of the three models, representing the error between the actual and predicted values. (**b**) displays the MAE prediction results of the three models, reflecting the average absolute error during the prediction process, which indicates the accuracy and stability of the models’ predictions.

**Figure 8 sensors-24-06857-f008:**
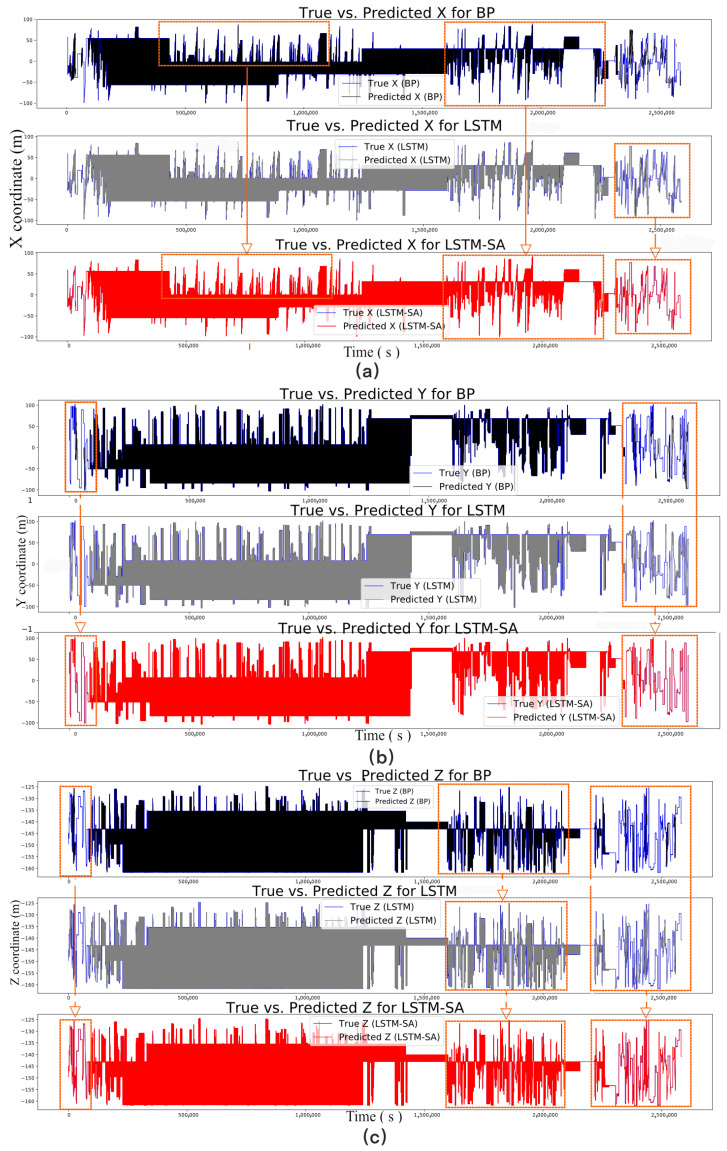
Time series predictions for the FAST feed cabin using the LSTM model, LSTM-SA model, and BP model. Subfigures (**a**–**c**) present the comparisons of the actual results along the X, Y, and Z axes, respectively.

**Figure 9 sensors-24-06857-f009:**
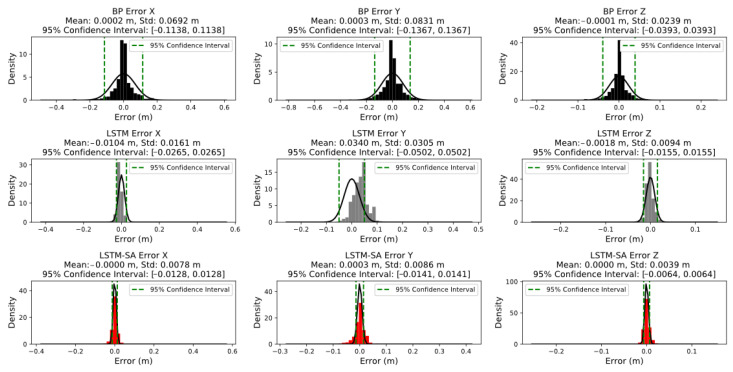
RMSE and MAE variations over time for BP, LSTM, and LSTM-SA models when predicting June 2023 data. The BP model exhibits the largest errors, with consistent deviations above 15 mm, while the LSTM model shows moderate improvement but still fails to meet the target. The LSTM-SA model demonstrates the best performance, with most errors below 15 mm, reduced maximum error, and more stable predictions.

**Figure 10 sensors-24-06857-f010:**
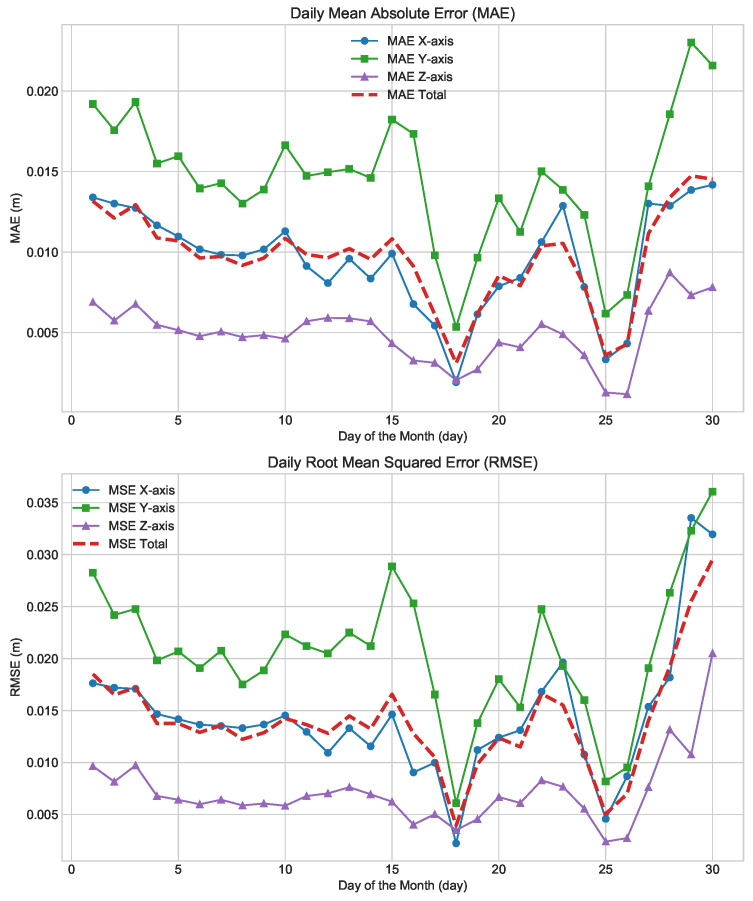
The image illustrates the daily performance of the LSTM-SA model on the X, Y, and Z axes in June, including RMSE and MAE. The green line represents the Y-axis, the blue line the X-axis, and the purple line the Z-axis. The red line indicates the average of the X, Y, and Z axes. Overall, the Z-axis performs the best, followed by the X-axis, with the Y-axis performing the worst. Temporally, the first five days show poorer performance, with errors gradually decreasing over time, but there is a rise in errors during the last five days.

**Table 1 sensors-24-06857-t001:** Observation duration and Pearson correlation analysis of feed cabin position prediction across different motion modes. The Pearson correlation analysis is based on the calculation between the cumulative observation duration for various observation modes in June 2023 and the RMSE of the LSTM-SA model at the same time points.

Observation Mode	Duration (s)	Correlation
MultiBeamOTF	47,496	0.338666
OnTheFlyMapping	16,674	0.302489
DriftWithAngle	68,400	0.235603
PhaseReferencing	0	NAN
Tracking	133,977	0.060534
OnOff	59,446	−0.001844
SnapShotCal	25,980	−0.026541
SnapShot	41,340	−0.028853
SwiftCalibration	144,520	−0.111182
MultiBeamCalibration	4800	−0.469986
DecDriftWithAngle	0	NAN
TrackingWithAngle	0	NAN

**Table 2 sensors-24-06857-t002:** Parameters and values during model training.

Parameter	Value
Dataset size	129.6 million entries
Time frame	January to May 2023
Batch size	64
Learning rate	0.0001 (initial)
Optimizer	AdamW
Training epochs	1500 or early stopping
Hidden units (LSTM)	256
Layers (LSTM)	8
Attention heads	2 (dual-head attention)
Loss function	Mean Squared Error (MSE)
Learning rate schedule	ReduceLROnPlateau
Training-validation split	Training: 80%, Validation: 20%
Validation strategy	Early stopping based on validation loss

## Data Availability

All input parameter for model computations are described in the manuscript. The code product that supports this work is publicly available from https://github.com/ShuaiPengGY/LSTM_SA, accessed on 21 October 2024. FAST observations can be requested at https://fast.bao.ac.cn/.

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
