# Peer review of "Study of Five-Hundred-Meter Aperture Spherical Telescope Feed Cabin Time-Series Prediction Studies Based on Long Short-Term Memory–Self-Attention"

_sensors, 2024, doi:10.3390/s24216857_

Round 1

Reviewer 1 Report

Comments and Suggestions for Authors

The study uses deep learning to predict FAST feed cabin time-series. The manuscript needs further revisions, and my opinions are as follows:

1.In the Abstract:FAST operational precision standard 15mm, this is accuracy the official threshold or the self-set accuracy of 15mm, We need to clarify the source of 15mm. If it is the official threshold set, please optimize the summary description.

2.In the1. Introductionpart: We need references to prove that 'FAST's current accuracy requirement of 15 millimeters.'

3. In the “Materials and Methods, why did the author choose the LSTM model? LSTM is a special type of recurrent neural network (RNN) proposed by Hochreiter and Schmidhuber in 1997. LSTM aims to solve the problem of gradient vanishing or exploding encountered by traditional RNNs when processing long sequence data. In the FAST feed cable prediction research, the author only used BP and LSTM as comparison models, and the model construction steps of the fusion model LSTM-SA need to be supplemented.

4. In 171~173: The parameters presented in the study are not comprehensive. The author should increase the training set, testing set, learning rate, and other parameters data.

5. 2.2.1 and 2.2.2 section: LSTM and SA are existing models that can simplify the introduction process and add a framework diagram of the model. The LSTM model framework diagram can refer to the following references:

Sherstinsky, A. (2020). Fundamentals of recurrent neural network (RNN) and long short-term memory (LSTM) network. Physica D: Nonlinear Phenomena, 404, 132306.

Zhao, Z., Chen, W., Wu, X., Chen, P. C., & Liu, J. (2017). LSTM network: a deep learning approach for shortterm traffic forecast. IET intelligent transport systems, 11(2), 68-75.

Pulver, A., & Lyu, S. (2017, May). LSTM with working memory. In 2017 International Joint Conference on Neural Networks (IJCNN) (pp. 845-851). IEEE.

6. In 146~222: there is no reference in the introduction of the model method in the manuscript. The introduction of the data source can be added to the website. Please optimize and supplement the corresponding literature throughout the manuscript.

7. In 346~350: MSE, RMSE, and MAE result decimal points should be kept to 4 or 5 digits. Please set them uniformly and include the unit after each result value, such as 0.00085m, 0.029m, and 0.195 m.

8. “3.2. Comparison and analysis”: this section mainly evaluates the quality of the model through MSE, MAE, R2and RMSE. Why is there no distribution of predicted results for FAST feed cable time series? The time series prediction results under different models need to be visualized, and MSE is the evaluation of the results.

9. In 512~515: The robustness statement of the LSTM-SA model is inaccurate. Robustness refers to the ability of a system to maintain its functional stability even when faced with changes in internal structure or external environment. LSTM-SA is limited to FAST feed cable time series prediction in the manuscript and lacks robustness. Please optimize the conclusion and draw a conclusion based on the content of the manuscript. Please verify the author's conclusion.

10. There is a serious lack of references (only 25), especially in the introduction of LSTM and SA models. The manuscript needs to add more references.

Reviewer 2 Report

Comments and Suggestions for Authors

The paper “Study of FAST feed cabin time-series prediction studies based on LSTM-Self-Attention” is devoted to the new Long Short-Term Memory-Self-Attention model, which allows increasing the operating time by reducing the influence of the environment.

The presented research on the prediction of the position of the five-hundred-meter aperture spherical telescope (FAST) feed cabin by integrating Long Short-Term Memory (LSTM) networks with self-attention (SA) mechanism confirms the better result compared to traditional methods, which include the use of back propagation (BP) models and standard LSTM models, in terms of accuracy, stability and robustness. The use of such models allows to expand the capabilities of using FAST in various conditions caused by environmental disturbances or interference from other signals, which in some cases leads to a complete stop of observations. The accuracy of the LSTM-SA model achieves a mean absolute error (MAE) of less than 10 mm and a root mean square error (RMSE) of approximately 12 mm, which is at the level of the standard FAST operating accuracy of 15 mm. This allows for highly sensitive astronomical observations to be carried out almost continuously with less dependence on environmental variability.

There are some points in the work that require corrections:

1.                 In Figure 1, the captions are hard to read. They need to be made larger.

2.                 The paper takes into account wind vibrations and provides an analysis of the prediction accuracy. However, it is not described in sufficient detail how exactly the use of the LSTM-SA prediction model allows preventing the measurement system from being turned off after the failure of devices based on the measurement data before the failure of devices. As indicated, such situations arise in the practical application of the feed cabin measurement system: GNSS receivers often fail due to interference from other satellite signals, and TS also often fail due to weather effects. In particular, it should be indicated under what weather conditions the use of the LSTM-SA model helps.

Round 2

Reviewer 1 Report

Comments and Suggestions for Authors

The author fix all the comments in the 1st round, now can be published now